# Zn-Doped Calcium Magnesium Phosphate Bone Cement Based on Struvite and Its Antibacterial Properties

**DOI:** 10.3390/ma16134824

**Published:** 2023-07-04

**Authors:** Polina A. Krokhicheva, Margarita A. Goldberg, Alexander S. Fomin, Dinara R. Khayrutdinova, Olga S. Antonova, Alexander S. Baikin, Aleksander V. Leonov, Ekaterina M. Merzlyak, Ivan V. Mikheev, Valentina A. Kirsanova, Irina K. Sviridova, Suraya A. Akhmedova, Natalia S. Sergeeva, Sergey M. Barinov, Vladimir S. Komlev

**Affiliations:** 1A.A. Baikov Institute of Metallurgy and Materials Science, Russian Academy of Sciences, Moscow 119334, Russia; mgoldberg@imet.ac.ru (M.A.G.); afomin@imet.ac.ru (A.S.F.); dvdr@list.ru (D.R.K.); oantonova@imet.ac.ru (O.S.A.); baikinas@mail.ru (A.S.B.); barinov_s@mail.ru (S.M.B.); 2Department of Chemistry, M.V. Lomonosov Moscow State University, Moscow 119991, Russia; avleonov49@gmail.com (A.V.L.); mikheev.ivan@gmail.com (I.V.M.); 3Department of Molecular Technologies, Pirogov Russian National Research Medical University, Moscow 117997, Russia; ekaterin99@mail.ru; 4P.A. Hertsen Moscow Oncology Research Institute—Branch of National Medical Research Radiological Centre Affiliated with Ministry of Health of Russian Federation, 2nd Botkinsky Pr. 3, Moscow 125284, Russia; kirik-57@mail.ru (V.A.K.); i.k.sviridova@yandex.ru (I.K.S.); tagieva58@mail.com (S.A.A.); prognoz.06@mail.ru (N.S.S.)

**Keywords:** calcium phosphate, magnesium phosphate, bone cement, antibacterial properties, cytocompatibility

## Abstract

The development of magnesium calcium phosphate bone cements (MCPCs) has garnered substantial attention. MCPCs are bioactive and biodegradable and have appropriate mechanical and antimicrobial properties for use in reconstructive surgery. In this study, the cement powders based on a (Ca + Mg)/P = 2 system doped with Zn^2+^ at 0.5 and 1.0 wt.% were obtained and investigated. After mixing with a cement liquid, the structural and phase composition, morphology, chemical structure, setting time, compressive strength, degradation behavior, solubility, antibacterial activities, and in vitro behavior of the cement materials were examined. A high compressive strength of 48 ± 5 MPa (mean ± SD) was achieved for the cement made from Zn^2+^ 1.0 wt.%-substituted powders. Zn^2+^ introduction led to antibacterial activity against *Staphylococcus aureus* and *Escherichia coli* strains, with an inhibition zone diameter of up to 8 mm. Biological assays confirmed that the developed cement is cytocompatible and promising as a potential bone substitute in reconstructive surgery.

## 1. Introduction

The tendency of growing life expectancy is observed all over the world. The number of age-related diseases and surgical operations has grown accordingly [1]. Among them, there are osteoporosis [2], injuries, and traumas of various types. The latter tends to increase in frequency in all age groups [3]. In addition to osteosarcoma, which is the most prevalent primary bone tumor mainly affecting children and young adults [4], often needs surgical interventions. Currently, surgery is commonly accompanied by antibiotic therapy [5] to reduce the risks of infections and inflammation. On the other hand, antibiotics may pose additional risks, related to the growing prevalence of drug-resistant bacteria [6] or other side effects [7]. Hence, alternative strategies concerning the application of antimicrobial materials, e.g., bone cement, are of great interest.

Bone cement was originally named “materials based on polymethyl methacrylate” and are currently used in clinical practice [8]. Nevertheless, these materials have several disadvantages, such as an exothermic setting reaction (which can lead to damage to adjacent tissues) and the need for revision surgery [9]. Therefore, a search for alternative materials is underway. In recent years, several resorbable calcium phosphate-based bone cements (CPCs) [10], magnesium phosphate-based bone cements (MPCs) [11], and MCPCs were proposed [12,13,14].

The antibacterial activity of such materials may be achieved in several ways. The first one is the impregnation of a cement matrix with antimicrobial agents. Although this solution is widespread and allows for the implementation of the concept of local drug delivery [15,16,17], several issues are still unresolved, such as the control of drug release kinetics and the above-mentioned growing antibiotic resistance of bacteria. Thus, an alternative approach devoid of these shortcomings and involving the introduction of various ions into the cement matrix is attractive.

The most common ion introduced into cement materials to achieve antimicrobial activity is silver. It has been demonstrated that the presence of silver ions can eliminate chronic osteomyelitis [18], and the introduction of Ag^+^ cations lead to greater mechanical strength and antibacterial effects against both gram-positive and gram-negative bacteria [19]. Nonetheless, it should be noted that silver is not an element that is present in the body in appreciable amounts and, therefore, there may be health risks associated with its toxicity [20]. At present, this question is not fully clarified. An alternative—zinc—is present in various organs and tissues and is involved in various metabolic processes [21]. The Zn^2+^ ion plays a critical role in bone growth by promoting osteoblast differentiation and inhibiting osteoclast differentiation [22]. Zinc in various compounds can exert antibacterial actions [23,24]. In addition, its presence can lead to further improvement of the mechanical characteristics of a material [25].

A newberyite-based MCPC described in our previous paper shows a neutral pH level (7–8), a setting time of 10–13 min, mechanical strength up to 22 ± 3 MPa, cytocompatibility, and acceptable matrix surface properties [13]. Improved mechanical properties, as well as cytocompatibility and antibacterial activity, have been achieved for newberyite struvite-based bone cements that are doped with silver ions [19]. On the other hand, regarding the controversial effects of silver, the development of Zn^2+^-doped MCPCs is expected to improve the antibacterial properties and osteogenic ability of MCPCs because a release of Zn^2+^ may enhance osteogenic differentiation of certain cells for accelerated bone regeneration. Based on the above, the present work is devoted to the development of the new Zn^2+^-doped MCPC and its setting processes, thermal behavior, and mechanical properties, as well as the characterization of dissolution in model liquids, antibacterial properties, and cytocompatibility in relation to the MG-63 cell line.

## 2. Materials and Methods

### 2.1. Synthesis of Cement Materials

The synthesis of cement powders is based on a (Ca + Mg)/P = 2.0 system with 40 mol.%. Mg substitution was carried out by the method of precipitation from an aqueous solution of salts according to the following equation described in our previous paper [13]:2.4Ca(NO_3_)_2_ + 1.6Mg(NO_3_)_2_ + 4(NH_4_)_2_HPO_4_ + 8NH_4_OH = Ca_2.4_Mg_1.6_O(PO_4_)_2_ + 16NH_4_NO_3_ + 6H_2_O(1)

Zn^2+^ ions were introduced into the synthesis mixture as solutions of a nitrate [Zn(NO_3_)_2_] in the amount of 0.5 and 1.0 wt.%. After solution mixing, the slurries were evaporated to maintain the predicted concentrations.

The cement powders were heat treated at 1150 °C followed by grinding in a planetary ball mill with zirconia balls in a dimethyl ketone medium for 20 min. Cement powders were obtained, and major phases that formed during the synthesis of magnesium-substituted calcium phosphate were MgO, Mg_3_Ca_3_(PO_4_)_4_, and Ca_2.589_Mg_0.411_(PO_4_)_2_, which were consistent with a CaO–P_2_O_5_–MgO phase diagram [26].

Cement materials were prepared by mixing a cement powder with a cement liquid in a 2:1 ratio under sterile conditions on a glass slide using a spatula. The cement liquid was synthesized on the basis of ammonium phosphate (NH_4_)_3_PO_4_ and magnesium hydrogen phosphate MgHPO_4_ at a controlled pH value of 4.6. Cement materials with the following names were obtained: MCPC, 0.5%Zn-MCPC, and 1.0%Zn-MCPC. For material characterization and mechanical tests, cement samples were prepared in cylindrical form with a diameter of 0.8 mm; after setting, the samples were hardened in a 100% humidity atmosphere for 24 h.

### 2.2. Characterization of the Materials

Chemical composition was investigated by atomic emission spectrometry with inductively coupled plasma (AES-ICP, Vista Pro, ICP Expert 4.0 Software). The uncertainty of the ICP results was 2%.

The granulometric composition of cement powders was examined by means of a laser particle analyzer FRITSCH Analysis 22. The phase composition of the materials was determined by X-ray phase analysis (XRD) (on a Shimadzu XRD-6000, Kyoto, Japan) using CuKα radiation in the 2θ range from 10° to 70° with a step of 0.02° using the ICDD database, PDF2. The phases were matched with the following cards: magnesium oxide (MgO: ICDD 77-2364), a magnesium-substituted whitlockite phase (Ca_2.589_Mg_0.411_(PO_4_)_2_: ICDD 87-1582), stanfieldite (Mg_3_Ca_3_(PO_4_)_4_: ICDD 73-1182), newberyite (MgHPO_4_·3H_2_O: ICDD 75-1714), and struvite (MgNH_4_PO_4_·6H_2_O: ICDD 77-2303). The uncertainty of the XRD results was 5%. The particle size of the cement powders, as well as the microstructure of set cement materials, were investigated by scanning electron microscopy (SEM) (Tescan Vega II, Brno, Czech Republic). An analysis of the SEM data was carried out using ImageJ software 1.52u based on the secant method for at least 50 line segments.

Fourier-transform infrared absorption (FTIR) spectra of the samples were obtained using the KBr technique on a Nicolet Avatar-330 FTIR spectrometer (Thermo Fisher Scientific, Nicolet, Waltham, MA, USA) from 4000 to 400 cm^−1^ to evaluate the functional groups of the samples.

The pH value of the solutions and temperature during the mixing of the cement paste was measured with the help of a Testo pH meter (Testo, Titisee-Neustadt, Germany).

The setting time of cement materials was determined by immersion of a 1.0 mm (400 g) Vicant needle in a sample according to ISO 9917 (2007) [27]. The pH values of the extracts of set cement samples were determined after 1, 7, and 28 days in model fluids—the Kokubo Simulated Body Fluid (SBF) [28] and Dulbecco’s phosphate-buffered saline without calcium and magnesium (DPBS) (Thermo Scientific, Waltham, MA, USA)—at a ratio of 0.2 g/mL at room temperature.

### 2.3. Mechanical Testing

An Instron 5581 uniaxial testing machine was employed to measure the compressive strength of the cement samples at a crosshead speed of 1 mm/min according to ASTM D695-91. Five samples of each composition were analyzed to determine compressive strength, and the results are reported as mean ± standard deviation (SD).

### 2.4. Dissolution Assays

To understand the degradation behavior and solubility of MCPC and 1.0%Zn-MCPC, samples in the form of 28-day set disks (6 mm in diameter and 2 mm in height) were soaked in the model fluids the SBF and DPBS at 37.0 °C at a *w*/*v* ratio of 0.2 g/mL and a surface area-to-volume ratio of 0.1 cm^−1^ for various periods in a closed system [29]. At predetermined time points (1, 3, 7, 14, 21, and 28 days), two samples were taken out of the solutions, frozen in a ULT Freezer MDF-60U50 (SANYO Electric Co., Ltd., Osaka, Osaka, Japan) at −70 °C, and lyophilized in LS-1000 Prointeh-bio before weight loss calculations, and one sample was dried at 60 °C for studying the effect of holding time on phase composition and microstructure upon dissolution.

### 2.5. Testing of Antibacterial Properties

Antibacterial activities of MCPC, 0.5%Zn-MCPC, and 1.0%Zn-MCPC crystals were assessed against the gram-positive bacterium *Staphylococcus aureus* (strain ATCC 6538) and the gram-negative bacterium *Escherichia coli* (strain XL1-Blue) in accordance with ISO 20645 [30]. Tested cement samples were applied to the surface of the agar. The results of antibacterial activity were recorded after 24 h. The size of a zone of growth inhibition of the microorganisms was determined by measuring the distance from the edge of a tested cement sample to the border of the growth of a microorganism around the tested sample, as well as the absence or presence of bacterial growth in the zone of contact of the sample with the agar medium.

### 2.6. Testing of Cytocompatibility

In vitro cytocompatibility of cement samples was evaluated by the MTT assay on a human osteosarcoma MG-63 cell line on the 1st and 3rd days of cell growth (Russian Collection of Cell Cultures, Institute of Cytology, Russian Academy of Sciences, St. Petersburg, Russia). Before the start of the in vitro assays, the cement samples were sterilized with γ-radiation at a dose of 18 kGy. Cement disk samples 5.0 mm in diameter and 3.0 mm high were placed into 96-well plates for cultivation (Corning Costar, New York, NY, USA) in triplicate at one 96-well plate per incubation period and were covered with a complete growth medium (CGM), which consisted of DMEM (PanEco, Moscow, Russia), 10% of fetal bovine serum (Cytiva, Traun, Austria), 60 mg/mL glutamine (PanEco, Moscow, Russia), 20 mM HEPES buffer (PanEco, Moscow, Russia), and 50 μg/mL gentamycin (PanEco, Moscow, Russia). Before that, in the 96-well plates, MG-63 cells were seeded at a density of 15,000 cells per well in 200 μL of the CGM. All the procedures were performed under sterile conditions at 37 °C in an atmosphere of humidified air that contained 5% CO_2_. Subsequently, the optical density (OD) of the formazan solution (the end product of the MTT reaction) was evaluated on a Multiscan FC spectrophotometer (Thermoscientific, Waltham, MA, USA) at a wavelength of 540 nm, and the calculation of the population of viable cells (PVC) in relation to the control (in %) was carried out according to the following formula:PVC = OD_exp_/OD_contr_ × 100 (%),
where OD is the value of the optical density of the formazan solution.

A sample of the cement was assumed to be cytocompatible at PVC ≥ 70%. The details of this method are described elsewhere [31].

## 3. Results

### 3.1. Characterization of the Cement Powders

Based on the AES-ICP results, Zn^2+^ concentrations in the cement powders were close to the calculated ones (Table 1).

According to the results of the XRD analysis, the main phases of Zn-free magnesium calcium phosphate were magnesium-substituted whitlockite [Ca_2.589_Mg_0.411_(PO_4_)_2_] in the amount of 42 wt.% and a stanfieldite phase of monoclinic configuration with unit cell parameters of a = 22.8449(2) Å, b = 9.9999(1) Å, and c = 17.0882(4) Å at 29 wt.%; the rest of the magnesium did not enter these phases and crystallized as MgO at 29 wt.% (Table 1). Subsequently, this cement powder (MCPC) was used as a control sample. The Zn^2+^-substituted materials showed the formation of whitlockite and stanfieldite phases and magnesium oxide, similar to the undoped 40 mol.% Mg cement powder (Figure 1). With the increase in the Zn^2+^ content of the cement, the quantity of the MgO phase noticeably increased.

The introduction of Zn^2+^ into the material led to nonsignificant changes (an increase) in unit cell parameters of the stanfieldite phase (a = 22.8274(1) Å, b = 10.0012(1) Å, c = 17.0847(4) Å). We can hypothesize that the substitution of Mg^2+^ ions with Zn^2+^ (the ionic radius of Zn^2+^ is 0.83 Å vs. 0.65 Å in Mg^2+^ [32]) followed by a rise of the MgO amount in the materials.

The particle size distribution after grinding the cement powders is presented in Table 1. The average particle size of the MCPC is about 20 μm. It was found that under the same grinding conditions, the addition of Zn^2+^ leads to an increase in particle size up to 36 μm at 1.0 wt.% doping (Figure 2).

Infrared (IR) spectra of the obtained powders are given in Figure 3. A broad phosphate band in the region of 1200–900 cm^−1^ and well-resolved bands of ν_3_(PO_4_) at 1118, 1034, 991, and 970 cm^−1^ and ν_1_(PO_4_) at 943 cm^−1^ indicate that the structure of phosphate phases formed successfully. ν_4_(PO_4_) bands at 601 and 557 cm^−1^, as well as ν_2_(PO_4_) at 513 cm^−1^, are also clearly visible in all the materials. A broad peak formed by three bands at 1935, 2008, and 2081 cm^−1^ is related to the C=O vibration from residual acetone after sample milling. In the absence of dopant ions, this region has a wide band at 419 cm^−1^ and a low-intensity band at 405 cm^−1^. In the presence of zinc, a narrower band is observed for the 0.5 wt.% substitution at 410 cm^−1^, and for the 1.0 wt.% substitution, a doublet is seen at 419 and 410 cm^−1^.

### 3.2. Characterization of the Cement Materials

According to the XRD analysis (Figure 1b), after mixing the cement powders with the cement liquid, the formation of struvite (MgNH_4_PO_4_∙6H_2_O) orthogonal syngony with unit cell parameters of a = 6.9614(2) Å, b = 6.1480(3) Å, and c = 11.2507(2) Å was observed; the preservation of a small amount of initial stanfieldite, magnesium-substituted whitlockite, and MgO were also detected (Figure 4). In contrast, cement materials doped with 0.5 wt.% Zn^2+^ showed the formation of both struvite (a = 6.9500(1) Å, b = 6.1426(2) Å, and c = 11.2166(1) Å) and newberyite phases orthogonal syngony with unit cell parameters of a = 10.2287(2) Å, b = 10.7026(3) Å, and c = 10.0355(2) Å (MgHPO_4_∙3H_2_O) (Table 2). It should be noted that the introduction of the zinc cations in the amount of 0.5 wt.% resulted in a slight increase in the amount in the newberyite phase. Cement material based on the powders doped with 1.0 wt.% Zn formed the struvite phase (a = 6.9497(3) Å, b = 6.1342(4) Å, abd c = 11.2075(2) Å) and newberyite phase (a = 10.2186(1) Å, b = 10.6936(1) Å, c = 10.0312(4) Å).

The IR spectra (Figure 4) contain bands related to PO_4_ vibrations at 1125, 1030, 1065, 990, and 975 cm^−1^ [ν_3_(PO_4_)], 943 cm^−1^ [ν_1_(PO_4_)], 602 and 556 cm^−1^ [ν_4_(PO_4_)], and 512 cm^−1^ [ν_2_(PO_4_)]. It should be pointed out that with the increase in the amount of the dopant ion, the intensity and resolution of these bands noticeably diminished.

Ammonium salt was present in the setting cement liquid, which gave rise to a new phase: struvite (NH_4_MgPO_4_∙6H_2_O). Its presence was confirmed by the XRD analysis. The IR spectra also show absorption related to the NH_4_ group vibrations in the region 1400–1710 cm^−1^, as well as in the region of 3600–3200 cm^−1^. In the latter, H-O-H vibrations of water contribute and are present both in the newly formed crystal hydrates (struvite and brushite) and in free form.

The preparation of a cement material is based on chemical reactions between compounds of the cement powder and cement liquid. When phosphate salts (NH_4_)_3_PO_4_ and MgHPO_4_ come into contact with the cement liquid, their molecules dissociate and, within a few seconds, the cement liquid is saturated with ions in accordance with the following reactions:MgHPO_4_(aq) → HPO_4_^2−^(aq) + Mg^2+^(aq)(2)
HPO_4_^2−^(aq) → PO_4_^3−^(aq) + H^+^(aq)(3)
(NH_4_)_3_PO_4_(aq) → PO_4_^3−^(aq) + 3NH_4_^+^(aq)(4)

Futher, MgO recrystallization into struvite occurs (Krokhicheva et al., 2023), and dissolved ions interact with the original phases of the cement powder. Due to the presence of Ca^2+^ at 60 mol.%, we assume the emergence of amorphous crystalline hydrate phase CaHPO_4_∙2H_2_O (brushite) via the following reaction:Ca_x_Mg_(3−x)_(PO_4_)_2_ + (NH_4_)_2_HPO_4_ + (15 − x)H_2_O →
2MgNH_4_PO_4_∙6H_2_O + (1 − x)MgHPO_4_·3H_2_O + xCaHPO_4_·2H_2_O(5)

The microstructure of the set cement samples was investigated, and it indicated the formation of new cementing phases on the surface (Figure 5). The MCPC sample is characterized by a dense poreless structure of the struvite phase. The introduction of Zn^2+^ ions led to a struvite crystal shape modification. The crystalline phase has geometric features, such as a honeycomb appearance with an average crystalline size of approximately 5–8 μm for 0.5%Zn-MCPC. On the surface of the Zn^2+^-doped cement samples, the emergence of the new cementing crystalline hydrate phase newberyite was registered. On the surface of the sample’s cracks, we noticed the newberyite phase in the BSE mode.

The MCPC samples are characterized by a compressive strength of about 24 ± 3 MPa. It was found that the introduction of Zn^2+^ cations enhanced the strength. The greatest strength, 48 ± 5 MPa, was observed for 1.0%Zn MCPC samples (Figure 6a). During the mixing of the cement paste, no noticeable thermal reaction was detectable (Figure 6b). Measurement of the pH value of the solutions based on distilled water containing a cement material showed a pH level close to neutral during exposure to the solution for more than 1 day. The pH value of the samples with 40 mol.% of Mg after 24 days of holding in distilled water was 7.2.

The setting time for the cement materials was approximately 4–7 min regardless of composition. A setting time shorter than 15 min is adequate for surgery [33]. Additionally, the temperature was measured during the setting process of the cement material (Figure 6b). It was shown that the maximum temperature was 28 °C, which was observed at the moment of mixing cement powder with cement liquid. The next process of cement setting during 5 min is shown by the tendency of decreasing temperature for all cement compositions.

### 3.3. In Vitro Dissolution Assays

Degradation kinetics are determined by weight loss and pH value, and the profile is plotted as a function of time. The solubility assay of the cement materials revealed that on the first day of the experiment, the mass loss of the samples did not exceed 6%, regardless of the type of mortar. At the initial stage of the dissolution process, the unreacted cement liquid was removed, and subsequent features of the behavior of the cement materials in the model liquids were suggestive of processes taking place on the surface of the material, where interactions proceeded between the dissolved Ca^2+^, Mg^2+^, Na^+^, OH^−^, HPO_4_^2−^, and PO_4_^3−^ ions. The greatest mass loss in the MCPC sample was 7% and happened on the 7th day of the experiment in the model liquids; on the 14th day, the solubility in DPBS was less than the SBF; however, by the 28th day, the mass loss reached 5%. The introduction of zinc into the MCPC affected the nature of the solubility of the samples, and mass losses in the model liquids diminished; however, by the 21st day, the loss reached 7%, and by the end of the experiment, the solubility of the samples declined.

Measurement of the pH value in the SBF solution of the cement materials at various time points showed a neutral level of about 6.8–7.3, which was typical for struvite-based MCPCs [34] (Figure 7b).

After 28 days of soaking, SEM data revealed the formation of a porous and loose surface with flower-like structures (radial–radiant aggregates) of the newly formed bobierrite phase [Mg_3_(PO_4_)_2_∙8H_2_O], confirmed by energy-dispersive X-ray (EDX) analysis as a magnesium phosphate phase in the SBF (Figure 8). A structure similar to the above-mentioned radial–radiant structure and composed of whisker-like particles is present in macroporous MPC obtained from amorphous magnesium phosphate cement powders after soaking in the SBF for 7 days [35].

The bobierrite phase was probably produced by reaction 6 (see below). MgO is not soluble in water in the absence of acidic components but can create a favorable basic environment for bobierrite synthesis via the following reaction:3Mg^2+^(aq) + 2PO_4_^3−^(aq) + 8H_2_O = Mg_3_(PO_4_)_2_∙8H_2_O(6)

According to XRD data, by the 14th day of the experiment, the newberyite that was registered in the Zn^2+^-doped materials was fully dissolved in the SBF. The emergence of the new bobierrite phase was detected on the surface of the MCPC and Zn^2+^-doped samples after 28 days of soaking.

### 3.4. Antibacterial Properties

The antibacterial assays were designed to assess possible antibacterial activities of the doped cement, namely, the ability to inhibit bacterial attachment to a sample’s surface and bacteriostatic action, i.e., the capacity to inhibit bacterial growth. The results are presented in Table 3

According to the contact method results, the pure MCPC sample exerted antibacterial activity toward *S. aureus* because, owing to its slow dissolution, this material was transformed into magnesium hydroxide, which released hydroxyl ions when it dissociated [36]. By contrast, the struvite-based MCPC does not inhibit the growth of another gram-positive bacterium, *S. sanguinis*. In our test, the Zn^2+^-doped samples produced inhibition zones for both *S. aureus* and *E. coli*. As the amount of Zn^2+^ increased, the inhibition areas enlarged, as reported earlier when examining CPC materials [37].

### 3.5. Cytocompatibility of the Bone Cement Samples

The results of this assay indicated the absence of acute toxicity of the new bone cement to the cultured cells in question (Figure 9). The findings made on the 3rd day were consistent with the cytocompatibility of the newly developed formulations having bactericidal properties because, in their presence, the value of fatty acids in the tested osteosarcoma cells was 77–87%, with the exception of the sample containing 1.0 wt.% of Zn (fatty acid value: 65.3%). The observed behavior may be attributed to the arising amorphous magnesium phosphate phases, which yielded rapid proliferation and differentiation of osteoblast-like cells in contrast to the crystalline phase.

From the results of the in vitro assays, it can be concluded that the proposed bone cement is nontoxic and cytocompatible.

## 4. Discussion

The aim of the present study was to determine the effects of Zn^2+^ ions introduced to MCPCs. According to the XRD analysis, it was revealed that initial powders are characterized by the presence of whitlockite, stanfieldite, and MgO phases. The greater amount of introduced Zn^2+^ raised the concentration of MgO because there was a large quantity of Mg^2+^ displaced by Zn^2+^ ions in the stanfieldite lattice. After mixing the initial cement powder with a cement liquid, the main cementing phase (struvite) crystallized. One of the key findings in this study was that during cementing, the thermal reaction did not proceed at all in the tested compositions of the cement.

It is well-known that MgO is one of the main phases in MPC, which reacts by recrystallizing during its interaction with a cement liquid [38]. Several factors affect the rate of struvite formation during the mixing of magnesium calcium phosphate with a cement liquid based on (NH_4_)_2_HPO_4_ [39]. The hydration of MPC is basically an exothermal acid base neutralization reaction. On the one hand, there is the reactivity of the initial MgO because its specific surface plays a key part in the kinetics of the reaction. On the other hand, the type of phosphate acid applied and its concentration determine the solution’s initial pH, which evolves in the course of the chemical reactions. There is a hypothesis that the successive dissolution of MgO rapidly increases the pH value of the system [40]. When the pH value is greater than 7, NH_4_MgPO_4_·6H_2_O starts to form needle-like crystals and generates a network. The solubility product constant K_sp_ of struvite is 2.5 × 10^−13^ at 25 °C, whereas the K_sp_ of other products is far higher than this; therefore, struvite is the main hydration product of MPC.

Based on the XRD data unit cell parameters, the cell volumes were calculated for MCPC and Zn^2+^-doped MCPC struvite and newberyite phases. The amount of struvite volume cell of the MCPC is 0.485 nm^3^, 0.5 wt% Zn is 0.479 nm^3^, and 1.0 wt% Zn is 0.478 nm^3^. It could be concluded that doping with Zn^2+^ leads to a decrease in cell volume of struvite phase. These data were obtained for the first time for MCPCs. In addition, newberyite cell volumes were determined for Zn^2+^-doped materials; it was found that the introduction of 1.0 wt% Zn^2+^ led to a decrease in cell volume from 1.096 nm^3^ to 1.099 nm^3^ for 0.5 wt% Zn^2+^. This relation could be explained by assuming cell distortion due to the differences in ionic radii of Zn^2+^ and Mg^2+^.

It was noteworthy that the main difference between Zn^2+^-doped and Zn^2+^-free MCPCs is the appearance of the newberyite phase during the interaction with the cement liquid, according to XRD data. Boistelle et al. [41] have noted that the pH value and the degree of supersaturation of a substance in a solution play a key role in phase crystallization. They investigated the reaction between MgSO_4_ and NH_4_H_2_PO_4_ in equimolar amounts and demonstrated that struvite is metastable at specific pH levels (pH < 6.2) and supersaturations. Struvite precipitates first and then dissolves, generating newberyite (MgHPO_4_·3H_2_O), which is stable at a low pH value.

As shown in our work, the introduction of Zn^2+^ into the MCPC cement increases mechanical strength by almost twofold. In Ref. [42], researchers demonstrated the effect of the introduction of ZnO nanoparticles at up to 0.2 wt.% on the mechanical strength of Portland cement (PO 42.5). It was found that with an increase in the concentration of zinc oxide from 0 to 0.2 wt.%, the strength went up and amounted to ~62 and 78 MPa, respectively, by the end of the experiment. By contrast, on the 7th day of the experiment, the strength of zinc-doped cement was ~50 MPa. The same tendency of increasing strength with the introduction of Zn ions into the matrix was documented in Ref. [43], where the strength of cement not doped with zinc was ~15.4 MPa on day 7, whereas, for cement doped with zinc (1.4 wt.% of Zn), it was 17.5 MPa. In the present work, the synthesized MCPC doped with 1 wt.% of Zn^2+^ had a strength of 48 ± 5 MPa, which is comparable to structural Portland cement and almost two times stronger than CPC. Such high strength of the MCPC may be due to the nascent newberyite phase [44]. A similar effect—the strengthening of magnesium phosphate cement by doping with 1 wt.% of ZnO resulting in a mechanical strength of 49.2 MPa—is explained by the authors of Ref. [45]. The incorporated ZnO not only acted as stress concentration points absorbing a large amount of energy and promoting shear strain of the MPC matrix under the influence of external stress but also served as an effective barrier to dislocation movement, and efficiently impeded crack extension.

As already mentioned in the introduction, the present authors have evaluated MCPCs doped with silver ions [19] but, given the high cost of silver as a metal, the present authors decided to try an alternative: a study on doping with zinc. Zinc is known to have an antibacterial effect. Ref. [43] suggests that the addition of 1.4 wt.% of Zn^2+^ into brushite cement gives antibacterial properties against *E. coli*, *E. faecium*, and *P. aeruginosa*. In the work, the inhibition zone of *E. coli* for the MCPC was 14 mm, whereas, for the MCPC doped with 1.0 wt.% of Zn^2+^, it was 2 mm. This phenomenon is most likely due to the fact that the solubility of calcium phosphate cement is higher, and consequently, the release of zinc ions is more intense. It should be emphasized that gram-negative bacteria are more susceptible to zinc ions than gram-positive ones due to a thicker cell wall. No antibacterial effect against *E. coli* was observed in the case of the MCPC, but for both Zn^2+^-doped materials under study, sufficient antibacterial activity was achieved. Thus, doping with Zn^2+^ contributed to the antibacterial properties of the newly developed MCPCs. In Ref. [37], the average release of zinc ions from CPC at 0.6 wt.% of Zn was 0.307001 mg/L, and at 1.2 wt.% of Zn, it was 0.107001 mg/L. On the other hand, the results of antibacterial assays showed an inhibitory effect against pathogenic *E. coli* for CPC containing 0.6 wt.% of Zn^2+^. In our work, we demonstrated high antibacterial activity against the bacterium *S. aureus* ATCC 6538 at a lower yield of zinc ions, less than 0.1 mg/L. In Ref. [46], the researchers were unable to detect any bactericidal effect of CPC doped with zinc against gram-positive bacterium *S. aureus* ATCC 25923, whereas in our work, a strong inhibitory effect was registered for all compositions of the cement.

It is known that Zn^2+^ is a biocompatible ion and is present in many tissues in the human body [21,47]. For instance, the introduction of zinc into brushite cement at 1.4 wt.% improves the viability of NCTC L929 cells by approximately 10% relative to pure cement; this finding can be explained by the positive influence of Zn^2+^ ions on the proliferation of fibroblasts. A probable reason is the fact that zinc plays an important role in various biological processes and participates in a number of metabolic functions necessary for cell growth and development [43]. The introduction of 5 wt.% of zinc into apatite cement is reported to result in significantly higher proliferation rates of human osteoblasts compared to pure apatite [48], and it is proven that the release of zinc does not have toxic effects on the environment. In our work, all MCPCs were found to be nontoxic to the MG-63 cell line. The observed slight decrease in cell viability can be ascribed to the formation of amorphous phases in the cement and higher solubility on the 1st day of the experiment (6% mass losses). In Ref. [49], it is demonstrated that in biphasic mixtures of amorphous β-tricalcium magnesium phosphate, the viability of human mesenchymal stem cells decreased on the 7th day of the experiment. Those authors explain this outcome is due to rapid proliferation and the confluence of cells, i.e., the fastest cell proliferation at the beginning of the experiment. Further proliferation is slower.

Therefore, the developed cement materials are potentially useful in reconstructive surgery. Additionally, according to our analysis of the literature, there is no information on the doping of MCPCs with zinc cations; therefore, this study is novel in the field of cement materials. Here, in a CaO–P_2_O_5_–MgO system at (Ca + Mg)/*p* = 2.0 and 40 mol.% Mg substitution, we created MCPCs that possess improved physicochemical, mechanical, and biological properties because of a combination of different concentrations of Zn.

## 5. Conclusions

The present investigation shows the development of MCPCs with antibacterial properties due to the introduction of Zn^2+^ into cement powders based on MgO, whitlockite, and stanfieldite phases. With the help of XRD and FTIR data, we documented the formation of a newberyite phase during the setting and hardening; this phase led to major alterations in the cement’s properties. During the setting process, a thermal reaction was not observed. We observed greater mass loss during the soaking, enhanced cell growth in the MTT assay, as well as antibacterial activity. The increase in Zn^2+^ concentration from 0.5% to 1.0% caused a pronounced improvement of mechanical properties of the cement and enlarged the inhibition zone in a disk diffusion test against *E. coli* and *S. aureus* and did not lead to noticeable toxicity toward MG-63 cells.

## Figures and Tables

**Figure 1 materials-16-04824-f001:**
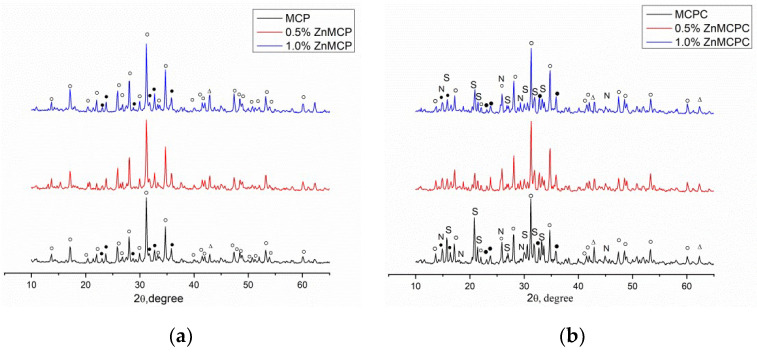
Diffraction patterns of (**a**) cement powders and (**b**) set cement. ○: whitlockite Ca_2.589_Mg_0.411_(PO_4_)_2_, ●: stanfieldite Mg_3_Ca_3_(PO_4_)_4_, S: struvite MgNH_4_PO_4_·6H_2_O, N: newberyite MgHPO_4_·3H_2_O, ∆: MgO.

**Figure 2 materials-16-04824-f002:**
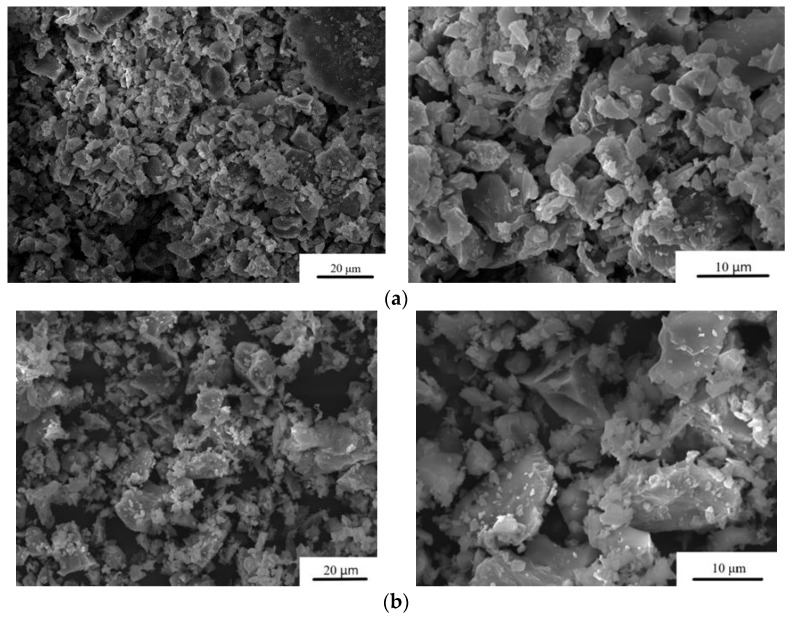
SEM images of the cement powders. (**a**) MCPC and (**b**) 1.0%Zn-MCPC in the SE mode.

**Figure 3 materials-16-04824-f003:**
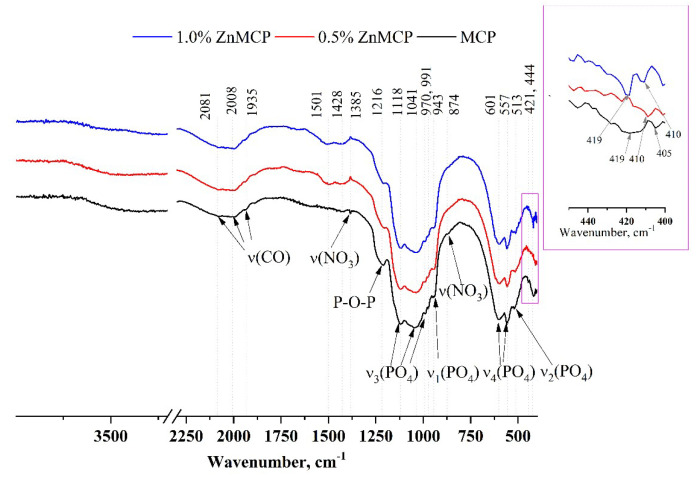
IR spectra of the powders after calcination and milling in acetone.

**Figure 4 materials-16-04824-f004:**
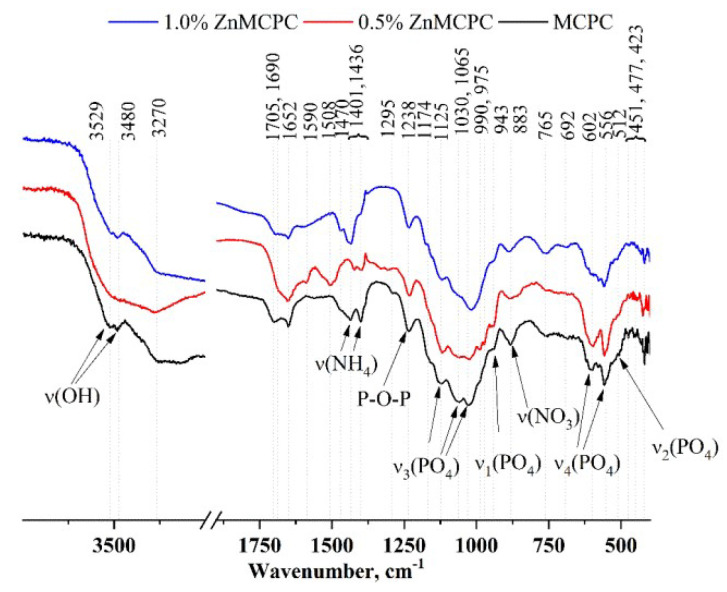
IR spectra of the cement samples after setting and hardening.

**Figure 5 materials-16-04824-f005:**
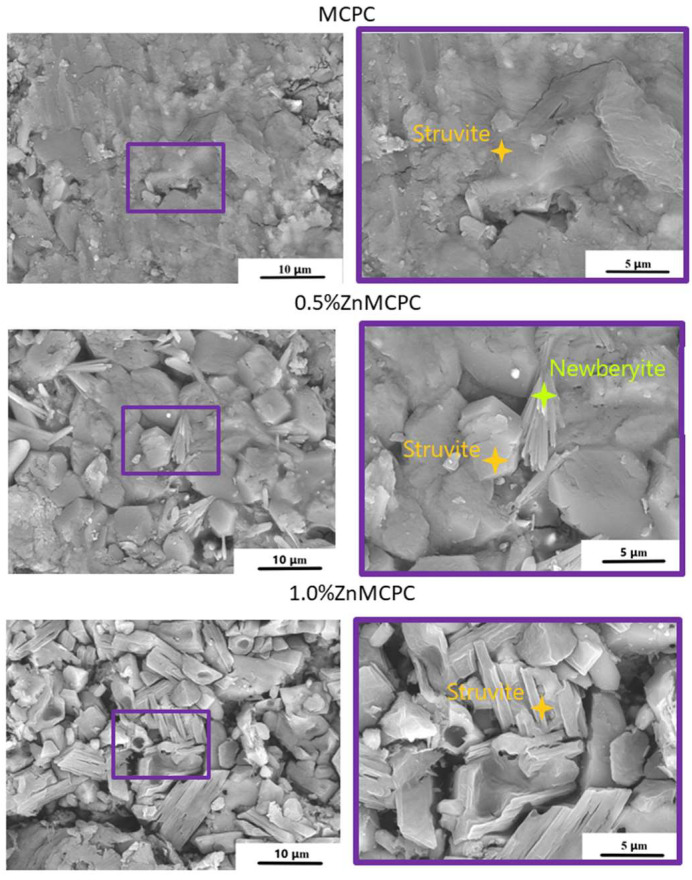
SEM images of the set cement containing 40 mol.% of Mg in the BSE mode (crack).

**Figure 6 materials-16-04824-f006:**
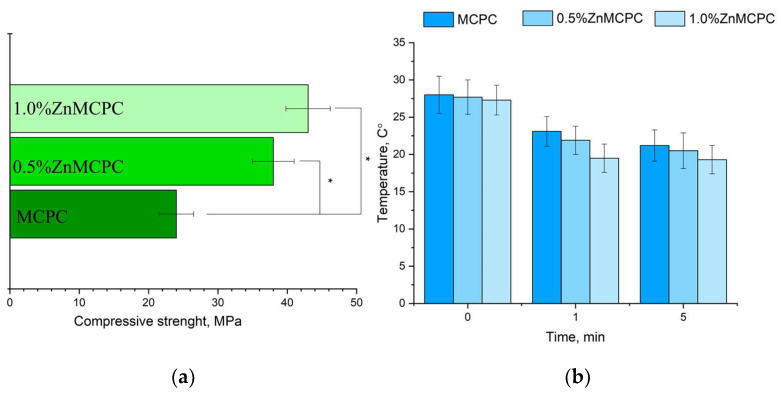
Results on the cement materials after mechanical testing for compressive strength (**a**); temperature measure during the setting process of cement materials (**b**). * Indicates significant differences compared between the Zn-doped samples and the control pure MCPC composition at *p* < 0.05 levels.

**Figure 7 materials-16-04824-f007:**
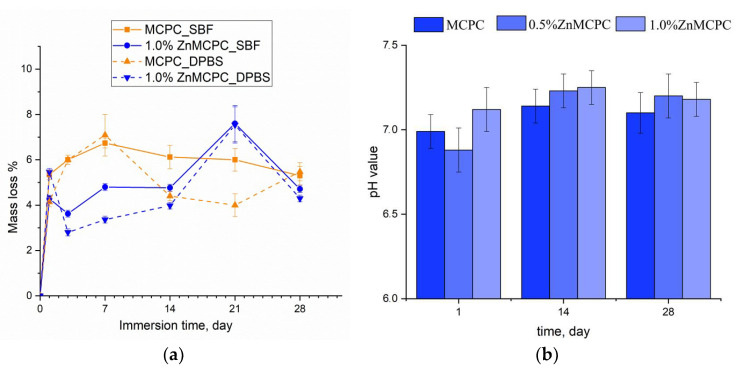
The dissolution assay of mass loss of the cement samples in the SBF and DPBS (**a**). The pH value in the SBF (**b**) throughout the experiment.

**Figure 8 materials-16-04824-f008:**
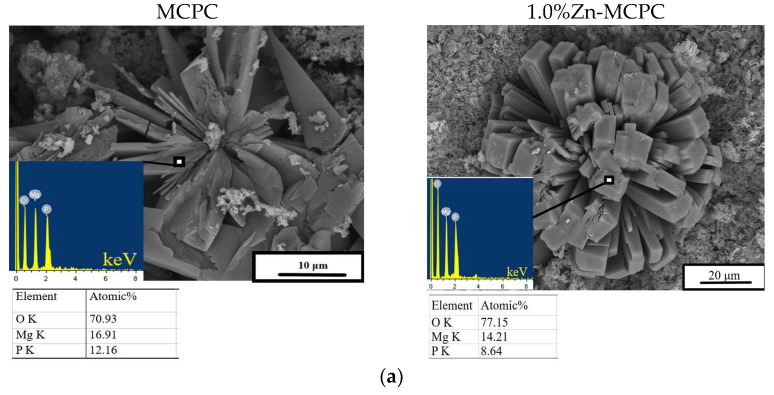
In vitro dissolution assays. Morphology (EDX data) of the cement materials’ surface after 28 days of soaking in the (**a**) SBF or (**b**) DPBS.

**Figure 9 materials-16-04824-f009:**
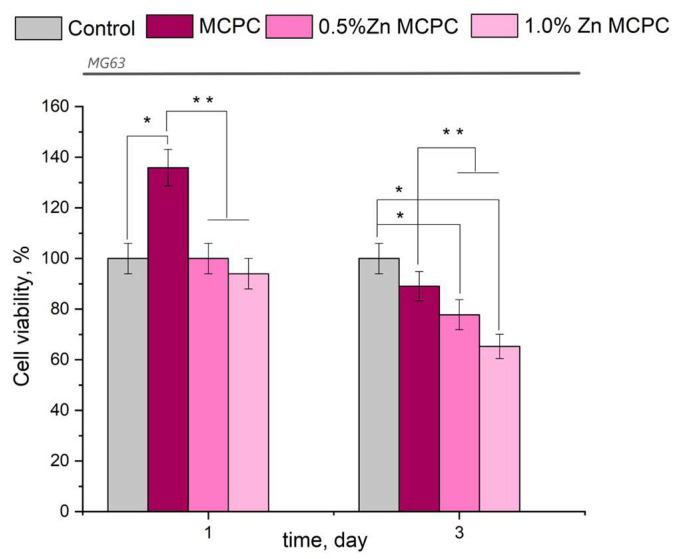
Viability of MG-63 cells during the cultivation on the bone cement samples or control samples (polystyrene). Data represent means ± SD (*n* = 2). * Indicates a significant difference compared to the control group at *p* < 0.05 levels. ** Denotes a statistically significant difference comparison among the groups (*p* < 0.05).

**Table 1 materials-16-04824-t001:** Phase composition of the cement powders and their particle size distribution.

Cement Powder Name	Phase Composition, wt.% *			Particle Size Distribution (μm)			Zn^2+^ Content, wt.% ** (ICP)
	Whitlockite Ca_2.589_Mg_0.411_(PO_4_)_2_	Stanfieldite Mg_3_Ca_3_(PO_4_)_4_	MgO	D10	D50	D90	
MCPC	42	29	29	1.0	7.9	21.6	0
0.5%Zn-MCPC	38	29	33	0.3	13.3	32.9	0.501
1.0%Zn-MCPC	39	22	39	0.5	18.0	36.4	0.911

* uncertainty 5%, ** uncertainty 2%.

**Table 2 materials-16-04824-t002:** Phase composition of the cement materials.

Set Cement Name	Phase Composition, wt.%				
	Whitlockite Ca_2.589_Mg_0.411_(PO_4_)_2_	Stanfieldite Mg_3_Ca_3_(PO_4_)_4_	MgO	Struvite MgNH_4_PO_4_ ·6H_2_O	NewberyiteMgHPO_4_ ·3H_2_O
MCPC	5	4	3	88	-
0.5%Zn-MCPC	8	6	3	58	25
1.0%Zn-MCPC	7	5	2	68	18

**Table 3 materials-16-04824-t003:** Inhibition zones of bacterial growth and an assessment of the efficacy of the antibacterial effect.

Samples	Concentration of Test Culture, 10^8^ Colony-Forming Units/mL (10 Units in BSS)		
	Growth Inhibition Zone, mm	Bacterial Growth under Sample	Assessment
*S. aureus* ATCC 6538			
MCPC	3	Completely inhibited	Sufficient effect
0.5%Zn-MCPC	7	Completely inhibited	Sufficient effect
1.0%Zn-MCPC	8	Completely inhibited	Sufficient effect
*E. coli* XL1-Blue			
MCPC	0	Growth was observed	Insufficient effect
0.5%Zn-MCPC	1	Completely inhibited	Sufficient effect
1.0%Zn-MCPC	2	Completely inhibited	Sufficient effect

## Data Availability

Not applicable.

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
