# Peer review of "Zn-Doped Calcium Magnesium Phosphate Bone Cement Based on Struvite and Its Antibacterial Properties"

_materials, 2023, doi:10.3390/ma16134824_

Round 1
Reviewer 1 Report
The work presented in the manuscript "Zn-Doped Calcium Magnesium Phosphate Bone Cements Based on Struvite and Their Antibacterial Properties" is very well conceived, presented and discussed with good connections to works on the same research topic. Minor revisions and suggestions are here reported:
- pag 3 line 1, Ca3Mg3(PO4)4: Authors report regularly stanfieldite as Mg3Ca3(PO4)4, thus change accordingly
- pag 3, paragraph "Characterization of the Materials": FTIR is not presented in this part even though spectra are part of the results. Please, add some lines. Moreover, please clarify at what hardening time FTIR and XRD analyses were performed. Is it also the same for mechanical testing? As regards phase composition, what is the uncertainty of the method?
- pag 3, paragraph "Mechanical testing", "dried": do you mean "hardened"/"set completely"?
- pag 4, paragraph "Testing of Cytocompatibility", line 10: cell line is barred, please remove it from the text. Similarly, check line 16. Check the editing of the Sentences from "Subsequently, optical density...". Check also the editing of the last two sentences of the same paragraph.
- Table 1: the table is not completely visible. Please add the unit of measurement for phase composition. Please report the same number of decimals for all the measurements in the Particle size distribution columns.
- pag 5, the Authors report that nonsignificant changes occurred in the unit cell parameters but only one value (from the standard stanfieldite? from the Zn-substituded analogous?) is reported for each parameter. I would suggest to clarify or remove the extimations.
- figure 1b: the quality of the image must be improved, as it the present form is blurry. The caption does not show the a/b letters. Please add.
- pag 5, "It was found that under the same grinding conditions...": even though it is clear by eye, please add a line on the methodology (i.e. image analysis by xxx software, based on xxx observations for each sample, after appropriate scaling). Was a statistical test performed as elsewere in the manuscript?
- pag 7, line 7: do the doped struvite and newberyite show any modification in the cell parameters? In case, does this occurrence entail any macroscopic effect (density, crystal matrix organization, variation in the amorphous phase total amount, behaviour under compression)?
- pag 7, line 1 after table 2, "Furthermore, according to XRD analyses...": the quality of the images (in particular 1b) does not support or give evidence of this claim. Could the authors change the image and give more detailed discussion? Furthermore, saving that the reduction of the resolution (and thus of the crystallinity of the phosphates) occurs with Zn increasing, the MCPC seems to show analogous or even enhanced behaviour. I would recommend a more detailed discussion if possible.
Finally, how many samples is each spectrum representative of? Has the decreasing trend of the resolution by Zn increment been verified by means of different samples?
- pag 7, line 2 from the bottom, "reflections": do you mean "absorptions" or were the IR spectra taken in reflectance mode? (in case, specify also in the Methods section)
- chemical reactions (2-4): please, remove the dots for spacing
- pag 8, line 4 from the bottom, "The introduction of Zn+2 ions led to a crystal shape modification": is it referred to struvite or newberyite?
- figure 6a: clarify the significance level of the asterisk and the double asterisk in the caption; the lines linking the bars to be compared are not correctly organized or there is an error in the use of asterisk or in the standard deviations dispayed over the bars. Otherwise how could be that the increase in compressive strength of 1%ZnMCPC is less significant than that of 0.5%ZnMCPC?
- pag 10, line 4, "It was shown...": please check the sentence.
- pag 10, line 9 from the bottom, "The greatest mass loss...": could the Authors suggest which phase is mainly dissolved at day 7, after the cement liquid?
- figure 9: please check the significance of the tested couples, similarly to what suggested for figure 6a. Moreover, consider to reorganizing the bars by the substrate (in couples) instead of by the time.
Could the Authors add a reference to evaluate threshold limits of cell viability to claim the absence of acute toxicity?
Minor revisions (already specified in the previous Comments section)
Author Response
Author’s Note to Reviewer 1
The work presented in the manuscript "Zn-Doped Calcium Magnesium Phosphate Bone Cements Based on Struvite and Their Antibacterial Properties" is very well conceived, presented and discussed with good connections to works on the same research topic. Minor revisions and suggestions are here reported:
Thank you very much for your detailed and interesting review. We appreciate your time.
- pag 3 line 1, Ca3Mg3(PO4)4: Authors report regularly stanfieldite as Mg3Ca3(PO4)4, thus change accordingly
Thank you. We corrected the text accordingly “…magnesium-substituted calcium phosphate were MgO, Mg3Ca3(PO4)4, and Ca2.589Mg0.411(PO4)2, consistently with a CaO–P2O5–MgO phase diagram [26].”
- pag 3, paragraph "Characterization of the Materials": FTIR is not presented in this part even though spectra are part of the results. Please, add some lines. Moreover, please clarify at what hardening time FTIR and XRD analyses were performed. Is it also the same for mechanical testing? As regards phase composition, what is the uncertainty of the method?
Thank you. We added completed information to the text accordingly ‘’.. Fourier-transform infrared absorption (FTIR) spectra of the samples were obtained using the KBr technique on a Nicolet Avatar-330 FTIR spectrometer (Thermo Fisher Scientific) from 4000 to 400 cm-1 to evaluate the functional groups of the samples’. In the paragraph Mechanical Testing added lines “After mechanical testing the of the cement materials X-ray and FTIR investigations were studied”. The uncertainty of the XRD and ICP results were 5% and 2%. The information was added in the manuscript in the chapter Characterization of the Materials.
- pag 3, paragraph "Mechanical testing", "dried": do you mean "hardened"/"set completely"?
Thank you for pointing this out. It will be corrected to use "hardened" instead of "dried". The corrected word has been added to the text.
- pag 4, paragraph "Testing of Cytocompatibility", line 10: cell line is barred, please remove it from the text. Similarly, check line 16. Check the editing of the Sentences from "Subsequently, optical density...". Check also the editing of the last two sentences of the same paragraph.
Thank you for your attention. All unnecessary words have been deleted.
- Table 1: the table is not completely visible. Please add the unit of measurement for phase composition. Please report the same number of decimals for all the measurements in the Particle size distribution columns.
Thank you. Measure unit of the X-ray is wt.%, information was added. The table was changed.
- pag 5, the Authors report that nonsignificant changes occurred in the unit cell parameters but only one value (from the standard stanfieldite? from the Zn-substituded analogous?) is reported for each parameter. I would suggest to clarify or remove the extimations.
Thank you. As the only difference was observed for lattice parameters of stanfeldite phase, the discussion of the whitlockite and MgO was out of the discussion object. Nevertheless, we added additional points in the text:” According to results of XRD analysis, the main phases of Zn-free magnesium calcium phosphate were magnesium-substituted whitlockite [Ca2.589Mg0.411(PO4)2] in the amount of 42 wt.% and stanfieldite phase of monoclinic configuration with unit cells parameters a=22.8449(2) Å, b= 9.999(1) Å , c= 17.0882(4) Å at 29 wt.% ; the rest of magnesium did not enter these phases and crystallized as MgO at 29 wt.% (Table 1, Figure 2).”
- figure 1b: the quality of the image must be improved, as it the present form is blurry. The caption does not show the a/b letters. Please add.
Thank you. The figure 6b was changed and information was added to the caption.
- pag 5, "It was found that under the same grinding conditions...": even though it is clear by eye, please add a line on the methodology (i.e. image analysis by xxx software, based on xxx observations for each sample, after appropriate scaling). Was a statistical test performed as elsewere in the manuscript?
Thank you. The sentence “It was found that under the same grinding conditions…” were used to draw attention than adding of Zn leads to grain size increased. The same grinding conditions described in the paragraph Materials and Methods “…followed by grinding in a planetary ball mill with zirconia balls in a dimethyl ketone medium for 20 min”.
The analysis of the SEM figures was performed by ImageJ software based on the secant method from no less than 50 line segments , the additional information was added to the Materials and Methods section: “Particle size of cement powders as well as the microstructure of set cement materials were investigated by scanning electron microscopy (SEM) (Tescan Vega II). Analysis of the SEM data was carried out using ImageJ software based on the secant method for at least 50 line segments.”
- pag 7, line 7: do the doped struvite and newberyite show any modification in the cell parameters? In case, does this occurrence entail any macroscopic effect (density, crystal matrix organization, variation in the amorphous phase total amount, behaviour under compression)?
Thank you. Information about unit cell parameters of the cementing phase were noticed and added to the text accordingly “According to the XRD analysis (Figure 1b), after the mixing the cement powders with the cement liquid, the formation of struvite (MgNH4PO4∙6H2O) orthogonal syngony with unit cell parameters of a= 6.9614(2) Å, b= 6.1480(3) Å, c= 11.2507(2) Å was observed; the preservation of a small amount of initial stanfieldite, magnesium-substituted whitlockite and MgO were also detected (Figure 4). In contrast, cement materials doped with 0.5 wt.% Zn2+showed the formation of both struvite (a=6.9500(1) Å, b= 6.1426(2) Å, c= 11.2166(1) Å) and newberyite phases orthogonal syngony with unit cell parameters of a=10.2287(2) Å, b= 10.7026(3) Å, c= 10.0355(2) Å (MgHPO4∙3H2O) (Table 2). It should be noted that the introduction of the zinc cations in the amount of 0.5 wt.% resulted in the slightly increase amount of the newberyite phase. Cement material based on the powders doped with 1.0 wt.% Zn formed struvite phase (a=6.9497(3) Å, b= 6.1342(4) Å, c= 11.2075(2) Å) and newberyite phase (a=10.2186(1) Å, b= 10.6936(1) Å, c= 10.0312(4) Å)”.
The unit cell values were calculated for Stanfieldite phase and Newberyite phase and some lines were added to the discussion part: “Based on the XRD data unit cell parameters, the cell volumes were calculated for MCPC and Zn2+-doped MCPC struvite and newberyite phases. The amount of struvite volume cell of MCPC is 0.485 nm3, for 0.5 wt% Zn is 0.479 nm3 and for 1.0 wt% Zn is 0.478 nm3, it could be concluded, that doping by Zn2+ leads to the decrease of cell volume of struvite phase. This data obtained for the first time for MCPC. In addition, newberyite cell volumes were determinated for Zn2+-doped materials, it was found that introduction of 1.0 wt% Zn2+ leads to decrease of the cell volume from 1.096 nm3 to 1.099 nm3 for 0.5 wt% Zn2+. This relation could be explained by assuming of the cell distortion due to of the differences in ionic radii of Zn2+ and Mg2”.
We did not observed any macroscopic effect due to high phases amount in the samples.
- pag 7, line 1 after table 2, "Furthermore, according to XRD analyses...": the quality of the images (in particular 1b) does not support or give evidence of this claim. Could the authors change the image and give more detailed discussion? Furthermore, saving that the reduction of the resolution (and thus of the crystallinity of the phosphates) occurs with Zn increasing, the MCPC seems to show analogous or even enhanced behaviour. I would recommend a more detailed discussion if possible.
Thank you for your comments. We have checked the spectra and agree with you that there is no noticeable amorphization. We have removed the sentence from the text.
Finally, how many samples is each spectrum representative of? Has the decreasing trend of the resolution by Zn increment been verified by means of different samples?
We conducted at list 3 investigation by XRD method of the samples of the same composition to avoid the erroneous conclusions.
- pag 7, line 2 from the bottom, "reflections": do you mean "absorptions" or were the IR spectra taken in reflectance mode? (in case, specify also in the Methods section)
Thank you for this comments, it is exactly means absorption, we corrected in the text: “The IR spectra also show absorption related to NH4 group vibrations in the region 1400–1710 cm-1 as well as in the region of 3600–3200 cm-1. To the latter, H-O-H vibrations of water contribute, which are present both in the newly formed crystal hydrates (struvite and brushite) and in free form”.
- chemical reactions (2-4): please, remove the dots for spacing
Thank you. Dots were removed.
13.pag 8, line 4 from the bottom, "The introduction of Zn+2 ions led to a crystal shape modification": is it referred to struvite or newberyite?
Thank you. The figure 5 was changed and phases were wrote.
14.figure 6a: clarify the significance level of the asterisk and the double asterisk in the caption; the lines linking the bars to be compared are not correctly organized or there is an error in the use of asterisk or in the standard deviations dispayed over the bars. Otherwise how could be that the increase in compressive strength of 1%ZnMCPC is less significant than that of 0.5%ZnMCPC?
Thank you. The significance level was corrected and figure 6(a) were changed. The description of the asterisk was added.
- pag 10, line 4, "It was shown...": please check the sentence.
Thank you. The sentence witch started with the words “It was shown…” located at the page 9. The sentence were rewrite according “It was shown, that maxima temperature28 C° observed in the moment of mixing cement powder with cement liquid, next process of setting cements up to 5 minutes demonstrated tendency in decreasing temperature for all cements composition.”
16.pag 10, line 9 from the bottom, "The greatest mass loss...": could the Authors suggest which phase is mainly dissolved at day 7, after the cement liquid?
Thank you. We assumed the dissolving Newberyite phase under 7th day of the experience based on XRD data of the 14th day.
17.figure 9: please check the significance of the tested couples, similarly to what suggested for
figure 6a. Moreover, consider to reorganizing the bars by the substrate (in couples) instead of by the time. Could the Authors add a reference to evaluate threshold limits of cell viability to claim the absence of acute toxicity?
Thank you. The significant differences was clarify, description was added “Data represent means ± SD (n = 2). * indicate significant difference compared to the control group at p < 0.05 levels. ** denote a statistically significant difference comparison among groups (p < 0.05).”
The absence of toxicity claims properly in the ISO 10993.5-9.

Reviewer 2 Report
The manuscript is very well written, the clarity of the results remarkable. Likewise, the results are relevant and presented in an excellent way. I consider that the article could be published in with minor corrections. Such as an implementation in the discussion of results, where citations of works are included in which the results of the present work can be compared.
Author Response
Author’s Note to Reviewer 2
The manuscript is very well written, the clarity of the results remarkable. Likewise, the results are relevant and presented in an excellent way. I consider that the article could be published in with minor corrections. Such as an implementation in the discussion of results, where citations of works are included in which the results of the present work can be compared.
Thank you very much for such positive review.

Reviewer 3 Report
The article “Zn-Doped Calcium Magnesium Phosphate Bone Cements Based on Struvite and Their Antibacterial Properties” present the experimental investigations of magnesium calcium phosphate bone cements (MCPCs) and their modified version doped with Zn2+ at 0.5 and 1.0 wt.%. After the mixing with a cement liquid the authors examined the several properties of new cements. Among other things, research was considered structural and phase composition, morphology, chemical structure, setting time, compressive strength, degradation behavior, solubility, antibacterial activities, and in vitro behavior of the cement materials.
The authors' summary regarding the obtained and tested samples seems to be correct and zinc-modified MCPCs seem to be a promising material as a potential bone substitute in reconstructive surgery.
My questions and suggestions:
Please modify the Fig.1 to be so clear as Fig. 3 or 4. Zoom in and properly crop the graph so that the markings of individual peaks in the spectrum are easily visible. There is too much empty space between graphs Fig.1.a and Fig.1.b.
Something went wrong with editing table 1 in the received pdf file of the manuscript. please edit this table better - some of the information is illegible.
This work is worthy of publication in Materials (publ. MDPI) with minor corrections listed as a comments/questions in attached pdf files and placed on the MDPI server in Report Review Form.
The text presented is correctly written in English. A few simple errors / typos I marked in the attached, checked pdf file.
Author Response
Author’s Note to Reviewer 3
- Please modify the Fig.1 to be so clear as Fig. 3 or 4. Zoom in and properly crop the graph so that the markings of individual peaks in the spectrum are easily visible. There is too much empty space between graphs Fig.1.a and Fig.1.b.
Thank you for attentiveness. All suggestions were accepted.
Something went wrong with editing table 1 in the received pdf file of the manuscript. please edit this table better - some of the information is illegible.
Thank you for attentiveness. The table was corrected.

Reviewer 4 Report
The work of Krokhicheva et al deals with the preparation and characterization of Zn-Doped Calcium Magnesium Phosphate Bone Cements. Structural, phase, compositional mechanical and biological properties were thoroughly studied in this work. Overall, the manuscript is well organized and easily followable with the effort to give novel findings. The presentation of the introduction, characterization methods, results, discussion and conclusions are well organized.
My comments to the manuscript are as follows:
Introduction
1) The Intro part properly described the literature review of the given topic, however the novelty and the clear contribution of this study should be better highlighted.
Materials and Methods
2) 2.6. Testing of Cytocompatibility
Please check if the actual version of text is present in the above paragraph. Some of the words are crossed out.
Results
3) The resolution of some Figures should be improved, such as Fig. 1a and b XRD diffraction patterns and the resolution of the scale bars in Fig. 5
Author Response
Author’s Note to Reviewer 4
The work of Krokhicheva et al deals with the preparation and characterization of Zn-Doped Calcium Magnesium Phosphate Bone Cements. Structural, phase, compositional mechanical and biological properties were thoroughly studied in this work. Overall, the manuscript is well organized and easily followable with the effort to give novel findings. The presentation of the introduction, characterization methods, results, discussion and conclusions are well organized.
Thank you very much for such positive review.
My comments to the manuscript are as follows:
Introduction
1) The Intro part properly described the literature review of the given topic, however the novelty and the clear contribution of this study should be better highlighted.
Thank you very much. We really hope that this study will expand the field of knowledge about MCPC, by your recommendation the novelty and scientific importance has been better highlighted and added to the Introduction: “Based on the above, the present work is devoted to the development of the new Zn2+ -doped MCPC and its setting processes, thermal behaviour, mechanical properties as well as characterisation of dissolution in model liquids, antibacterial properties and cytocompatibility in relation to the MG-63 cell line».
Materials and Methods
2) 2.6. Testing of Cytocompatibility
Please check if the actual version of text is present in the above paragraph. Some of the words are crossed out.
Thank you. All remarks were vanished.
Results
3) The resolution of some Figures should be improved, such as Fig. 1a and b XRD diffraction patterns and the resolution of the scale bars in Fig. 5
Thank you. The resolution of figures was improved.
